# Do Older Adults with Parent(s) Alive Experience Higher Psychological Pain and Suicidal Ideation? A Cross-Sectional Study in China

**DOI:** 10.3390/ijerph17176399

**Published:** 2020-09-02

**Authors:** Ying Yang, Shizhen Wang, Borui Hu, Jinwei Hao, Runhu Hu, Yinling Zhou, Zongfu Mao

**Affiliations:** 1School of Health Sciences, Wuhan University, Wuhan 430071, China; yangying@whu.edu.cn (Y.Y.); 2019203050022@whu.edu.cn (S.W.); 2018203050038@whu.edu.cn (B.H.); hjnwei@whu.edu.cn (J.H.); hurunhu@whu.edu.cn (R.H.); 2Global Health Institute, Wuhan University, Wuhan 430071, China

**Keywords:** psychological pain, suicidal ideation, Chinese elderly, adult children

## Abstract

Elderly mental health promotion is an important task in the current “Healthy China Action”. This study aims to (a) clarify the psychological pain and suicidal ideation of the Chinese elderly with different parental states, (b) examine the associated factors of psychological pain and suicidal ideation, and (c) examine the relationship between psychological pain and suicidal ideation. A sample of 4622 adults aged 60 years and older were included in this study, from the China’s Health-related Quality of Life Survey for Older Adults 2018. Participants with both parents alive demonstrated the heaviest psychological pain, and those with one parent alive observed significantly lowest psychological pain and suicidal ideation. Participants who were single, divorced, or widowed, live in rural areas, had higher education level, had lower family income, suffered from two or more chronic diseases, and had no self-care ability were more likely to experience psychological pain and suicidal ideation. In addition, higher psychological pain significantly associated with the occurrence of suicidal ideation. In China, much more attention should be paid to the mental health condition of the elderly, especially for those with both parents alive. Moreover, the associated factors above should be considered to develop targeted health interventions.

## 1. Introduction

Globally, the share of the population aged 65 years or older increased from 6% in 1990 to 9% in 2019 [1]. By the end of 2018, there were 249 million elderly residents aged 60 years and older in China, accounting for 17.9% of the total Chinese population [2]. This number is projected to reach 487 million (34.9%) by 2050. Accordingly, the elderly care needs and burden have increased dramatically [3,4]. In China, family-supporting constitutes the main channel of eldercare, and adult children play a pivotal role in home care for their parents [5,6]. Thus, they are more likely to experience severe physical, psychological, social, and economic burdens [7,8,9,10]. Lou et al. [9] reported approximately one-third of adult children caregivers for elderly people suffered from psychological distress in Hong Kong.

In China, filial piety (*xiao*) has dual impact on the elderly care. Filial piety, the core pillar of Confucian ethics, rests on children being respectful, obedient, and obligated to provide adequate care and support for their older parents both financially and emotionally. “Its demands range from material to emotional requirements, such as support, memorializing, attendance, deference, compliance, respect, and love” [11]. While filial piety culture might be reciprocal and promote good family relations [11], it also has as a consequence the elderly’s heavy reliance on their adult children [6,9,12] and affects the caregiving process and the caregivers’ well-being [13,14]. Particularly, in some areas, people believe filial piety means to obey parents’ wishes unconditionally, which brings much more pressure to adult children [15]. However, few studies have explored the mental health condition of caregivers who themselves are elderly.

Under cultural expectations and constraints, adult children of older age not only face their own aging and illness, but also might shoulder the burden of caring for their older parents, as well as other family members [16]. For elderly individuals with parent(s) alive, the difficulties they faced might be quite more serious. Firstly, they themselves might be exposed to lack of care and have unmet health needs, especially for mental health, emotional support and social relationships, healthy diet and exercise, and personal time [16,17]. Secondly, they might be expected to be the ultimate financial provider [18], suffer a high social burden and poor quality of life, as well as great psychological stress (anxiety, anger, etc.) [16,19], especially if their parents are in poor health condition. Thus, we considered that older people with parent(s) alive might be more likely to suffer from psychological problems in China.

In July 2019, the Chinese government implemented “Elderly Health Promotion Action” and “Mental Health Promotion Action” [20] as part of the “Healthy China Action (2019–2030)”. To clarify the above issue, and to provide references for the development of targeted interventions towards elderly people’s mental health in China, we conducted this exploratory study. This study divided elderly participants into three categories (neither parent alive, one parent alive, and both parents alive) according to their parental state and made an attempt to (a) assess the psychological pain and suicidal ideation of the Chinese elderly with different parental states, (b) determine the associated factors of psychological pain and suicidal ideation, and (c) examine the relationship between psychological pain and suicidal ideation.

## 2. Materials and Methods

### 2.1. Data Collection and Participants

This study used data from China’s Health-related Quality of Life Survey for Older Adults 2018 (CHRQLS-OA 2018). The survey employed a cross-sectional design and collected data using self-report questionnaires. In China, the CHRQLS-OA 2018 was organized and conducted by the Global Health Institute of Wuhan University during the Spring Festival in 2018. The survey aimed to collect data on the socio-ecological factors and health status of the elderly in China. All participants were aged 60 years and older. Using a convenience sampling strategy, a general database containing 5442 valid samples was finally established. The survey was conducted both online and offline during the Spring Festival, when the population is most evenly distributed in China. The response rate for the offline survey was 85.26%.

In this study, we excluded subjects with missing information on parental state, psychological pain, suicidal ideation, and demographic characteristics (*n* = 780). Finally, 4622 individuals aged 60 years and older were included in the analysis.

### 2.2. Measures

#### 2.2.1. Demographic Information

Participants’ socio-demographic characteristics were collected, including age, gender, marital status, place of residence, education level, family per-capita annual income (CHY), occupation before retirement, number of chronic diseases, and self-care ability.

In addition, participants’ parental state was investigated by the question “Is your father/mother alive: both parents alive, only father alive, only mother alive, or neither parent alive?” The parental state was divided into three categories in this study: (1) “neither parent alive”, i.e., both the participant’s father and mother are dead, (2) “one parent alive”, i.e., one of the participant’s parents is dead, (3) “both parents alive”, i.e., both the participant’s father and mother are alive.

#### 2.2.2. Psychological Pain

Psychological pain was assessed using the Chinese version of the Psychache Scale (PAS). The PAS was originally developed by Holden et al. [21] in 2001 and was translated into Chinese version by Yang and Chen [22] in 2017. It consists of 13 items utilizing a five-point Likert scale. Nine of the items reflecting pain frequency scored 1–5 from *never* to *always*, and four more items reflecting pain intensity scored 1–5 from *strongly disagree* to *strongly agree*. The sum-score ranges from 13 to 65, with higher scores reflecting heavier psychological pain. The Chinese version of the PAS was well validated in the Chinese population [23]. The Cronbach’s alpha of the Chinese version PAS is 0.97 in our sample and no inter-item correlation lower than 0.57.

#### 2.2.3. Suicidal Ideation

Three structured questions were adopted to assess suicidal ideation. Participants were asked to rate (1) wish to die (0 = none; 1 = mild; 2 = moderate to severe), (2) wish to live (0 = moderate to severe; 1 = mild; 2 = none), and (3) reasons to live/die (0 = more for living; 1 = equal; 2 = more for dying). The ratings of the three questions were then summed to yield a total score (0 to 6). Participants who reported a total score of ≥ 1 were considered as having suicidal ideation. Rather than asking participants directly about their wish to die by suicide or to kill themselves, we applied the three structured questions to indirectly assess the suicide risk of the elderly, which is milder and can potentially improve the respondents’ cooperation. Similar assessment has been widely applied in previous studies, for example, the “wish to die” criterion features among initial questions to access suicidal risk in the Columbia Suicide Severity Rating scale [24] and in the MINI Neuropsychiatric interview [25]. The three structured questions exhibited good internal consistency reliability in the present sample with Cronbach’s alpha = 0.77 and no inter-item correlation lower than 0.46.

### 2.3. Statistical Analysis

Data analyses were carried out by using the IBM SPSS version 23.0 for Windows (IBM Corporation, Armonk, NY, USA). A two-sided *p*-value < 0.05 was considered as statistically significant.

Data analysis was performed in four steps. First, descriptive statistics were conducted including means and standard deviations (SD) (continuous variables), frequency and percentage (categorical variables). Second, one-way ANOVA and chi-square test were used to examine the differences of psychological pain and suicidal ideation in different parental state groups.

Third, regression analyses were applied to examine the associated factors of psychological pain and suicidal ideation. Before constructing regression models, univariate analyses, including *t*-test, one-way ANOVA, Pearson correlation analysis, and chi-square test, were conducted to select descriptors as independent variables for regression models. Variables with *p*-value <0.1 were included for multivariable analysis. In this study, all the 10 socio-demographic characteristics listed in Table 1 demonstrated a *p*-value of less than 0.1 in the univariate analyses, and thus, they were all included in the regression models. Multiple linear stepwise regression method was used to test the associated factors of psychological pain. The total score of psychological pain was set as the dependent variable, and the model selection was automated. The factors associated with suicidal ideation were identified using binary logistic regression analysis. The condition of suicidal ideation (yes or no) was set as the dependent variable. All predictor variables entered into the same model, and the model selection was automated. Variables were excluded if they exceeded the tolerance level for multicollinearity.

Finally, the relationship between suicidal ideation and psychological pain was examined via Kendall’s tau-b correlation coefficients and logistic regression analysis. In the regression model, the condition of suicidal ideation (yes or no) was set as the dependent variable. The score of psychological pain was set as independent variable. Demographic characteristics were adjusted in the model as confounder factors.

### 2.4. Ethical Statements

The study protocol was reviewed and approved by the Institutional Review Board of School of Health Sciences and Faculty of Medical Sciences, Wuhan University (IRB No. 2019YF2050). Informed consent information was included with each questionnaire and introduced before the surveys. Surveys were only conducted if subjects were fully informed of the content and aim of this research project and agreed to participate.

## 3. Results

### 3.1. General Information

A total of 4622 elderly participants with an average age of 71.2 ± 7.8 were involved in this study. There were 73.9% (*n* = 3414) of the participants with neither parent alive, 17.0% (*n* = 788) with one parent alive, and 9.1% (*n* = 420) with both parents alive. Of the participants, 49.4% (*n* = 2284) were male, 67.7% (*n* = 3129) were married, 60.7% (*n* = 2805) lived in rural areas, 66.6% (*n* = 3078) received education level of primary school or below, 35.5% (*n* = 1639) reported family per-capita annual income of less than 15,000 CHY, 57.5% (*n* = 2657) used to be farmer or worker before retirement, 52.0% (*n* = 2403) suffered from one or more chronic diseases, 70.6% (*n* = 3265) had fully self-care ability. The detailed demographic information of the study sample is listed in Table 1.

### 3.2. Psychological Pain and Suicidal Ideation

Table 2 demonstrates the scoring results of psychological pain and suicidal ideation in participants with different parental states. The average score for psychological pain, pain frequency, and pain intensity were 22.1 ± 9.5, 14.8 ± 6.6, and 7.3 ± 3.4, respectively. The ratio for reporting suicidal ideation was 44.6%. Based on results of univariate analysis and pairwise comparison, participants with one parent alive exhibited the lowest psychological pain (*F* = 33.27, *p* < 0.001, *η^2^* = 0.014), pain frequency (*F* = 28.71, *p* < 0.001, *η^2^* = 0.012), and pain intensity (*F* = 34.18, *p* < 0.001, *η^2^* = 0.015), as well as the lowest ratio of suicidal ideation (28.8%) (*χ^2^* = 95.80, *p* < 0.001). For PAS frequency, pairwise comparison also detected significant differences between participants with neither parent alive and with both parents alive, that is, psychological pain frequency of the participants followed a trend of “one alive” < “neither alive” < “both alive”.

### 3.3. Factors Associated with Psychological Pain and Suicidal Ideation

Table 3 presents the results of multiple linear regression for psychological pain. When compared to participants with neither parent alive, those with one parent alive reported significantly lower psychological pain score (*B* = −0.95, *p* = 0.013), those with both parents alive had significantly higher score (*B* = 1.47, *p* = 0.002). Besides, participants who were older (*B* = −0.06, *p* = 0.003), were male (*B* = −0.82, *p* = 0.003), were married (*B* = −1.02, *p* = 0.001), had higher level of family income (*B* = −1.68, *p* < 0.001), and had fully self-care ability (*B* = −3.73, *p* < 0.001) were detected with lower psychological pain. Those who lived in rural areas (*B* = 1.30, *p* < 0.001), had higher education level (*B* = 0.55, *p* = 0.001), and suffered from ≥2 chronic diseases (*B* = 2.35, *p* < 0.001) exhibited higher psychological pain. The model showed good fitness (*R^2^* = 0.122, adjusted *R^2^* = 0.120, *F* = 63.61, *p* < 0.05), and the above variables explained 12.2% of the variance in psychological pain.

Table 4 presents the results of binary logistic regression for suicidal ideation. When compared to participants with neither parent alive, those with one parent alive reported significantly lower risk of suicidal ideation (*OR* = 0.72, 95% *CI* = 0.59–0.87). In addition, the results suggested significantly lower risk of suicidal ideation among participants who were married (*OR* = 0.51, 95% *CI* = 0.49–0.66), had higher level of family income (*OR* = 0.69, 95% *CI* = 0.64–0.73), and had full self-care ability (*OR* = 0.50, 95% *CI* = 0.43–0.58). Significantly higher risk of suicidal ideation was discovered among those who had higher education level (*OR* = 1.28, 95% *CI* = 1.18–1.38), suffered from one (*OR* = 1.36, 95% *CI* = 1.17–1.59) or ≥ 2 chronic diseases (*OR* = 1.92, 95% *CI* = 1.65–2.25). Hosmer-Lemeshow test indicated that the model had good fitness (*χ^2^* = 28.13, *p* > 0.05).

### 3.4. Association between Psychological Pain and Suicidal Ideation

Kendall’s tau-b coefficients and *ORs* with 95% CI were computed to clarify the association between psychological pain and suicidal ideation (Table 5). Significant correlations were detected between suicidal ideation and psychological pain (*r* = 0.44, *p* < 0.001), pain frequency (*r* = 0.42, *p* < 0.001), pain intensity (*r* = 0.43, *p* < 0.001). After adjusted for socio-demographic characteristics, psychological pain (*OR* = 1.16, 95% *CI* = 1.14–1.17), pain frequency (*OR* = 1.22, 95% *CI* = 1.20–1.24), and pain intensity (*OR* = 1.40, 95% *CI* = 1.37–1.44), all significantly associated with suicidal ideation. Hosmer–Lemeshow tests indicated good model fitness for psychological pain (*χ^2^* = 9.98, *p* > 0.05), pain frequency (*χ^2^* = 12.85, *p* > 0.05), and pain intensity (*χ^2^* = 11.37, *p* > 0.05).

## 4. Discussion

This study found Chinese elderly’s psychological pain and suicidal ideation differed by their parental states. Elderly individuals with both parents alive experienced the heaviest psychological pain, while those with one parent alive reported the lowest psychological pain and suicidal ideation. Marital status, place of residence, education level, family income, number of chronic diseases, and self-care ability were detected to be associated with psychological pain and suicidal ideation. Moreover, the severity of psychological pain significantly associated with the risk of suicidal ideation.

Psychological pain, called mental pain or psychache, is a lasting, unpleasant and unsustainable feeling characterized by a perception of inability or deficiency of the self [26]. We detected an average score of 22.1 ± 9.5 for psychological pain in Chinese elderly population, in line with the results reported by Holden et al. [21] in undergraduate students. In this study, 34.3% of the elderly participants declared a “wish to die”, which was higher than previous results assessed via the same question in American Facebook users (23.2%) [27], Portuguese undergraduate students (18.7%) [28], and Korean elderly people (14.5%) [29]. The ratio of reporting overall suicidal ideation was 44.6% in this study, which was much higher than several previous research found in Chinese elderly population (2.8% to 7.1%) [30,31,32,33]. The differences might be due to the distinction of measuring instruments of suicidal ideation. It is worth noting that this study indirectly assessed the elderly’s suicide risk through three structured questions, rather than directly asking about their suicidal ideation. Therefore, it is possible this study overestimated the prevalence of suicide ideation.

This study divided elderly participants into three categories (neither parent alive, one parent alive, and both parents alive) according to their parental state. For psychological pain severity, the obvious trend of “both alive” > “neither alive” > “one alive” was observed among the three groups. However, it should be noticed that parental state might only make a small contribution to the change of psychological pain, for the *η*^2^ of ANOVA tests ranges from 0.012 to 0.015 in this study. As for suicidal ideation, elderly individuals with one parent alive exhibited less risk than those with neither parent alive or both parents alive. In China, under the constraints of traditional Chinese culture of filial piety, elderly individuals with both parents alive might have to spend double time, energy, and money on taking care of their father and mother, especially if either of their parents suffers from diseases; thus, they presented more prominent psychological distress [6,15]. As for elderly participants without parents alive, since they are relieved from the burden of caring for their parents, they present less suicidal ideation. However, compared to participants with one parent alive, they do not benefit from a good parent–child relationship. This finding might affirm the continued importance of cultural norms of elderly care. Indeed, the elderly with one parent alive suffered less pressure for taking care of their parents, and could experience more family belongingness, especially domestic affection, companionship, support, etc., from their parent(s). This might be the reason why this group of elderly participants reported the least psychological pain and suicidal ideation. The role of family factors should not be ignored, Deng and John [34] found that the elderly’s mental health could be promoted by means of improving family belongingness. When it comes to the mental health management of older people, focusing on family factors interventions might make some sense.

We found that elderly participants with poorer individual health (suffered chronic disease, had no fully self-care ability) were more likely to experience psychological pain and suicidal ideation, which was consistent with previous findings [31,32,33,35]. The elderly with poor health status generally faced more problems regarding their own aging and illness, thus resulting in heavier psychological pressure.

Elderly participants who were married demonstrated substantial lower psychological pain and suicidal ideation than those divorced, widowed, or never married, in this study, which was in accordance with Yu et al. [32]. Previous studies have supported the effects of high social support and good family relationship on lowering suicidal ideation of older adults [36,37]. Elderly individuals with a partner could perceive spousal support, and thus their risk of psychological problems may decline. It was recommended that the mental health of older adults widowed or living alone should be noticed more.

In this study, elderly participants with lower family income demonstrated more severe psychological pain and suicidal ideation, and those living in rural areas also experienced greater psychological pain. Both living in rural areas and low family income represent low socioeconomic status, and thus, this finding might suggest that the elderly with low socioeconomic status were more likely to catch psychological problems. Similar results were reported by Wei et al. [31] and Yu et al. [32] in Chinese older adults.

We also discovered an interesting result that higher education level was associated with the increasing psychological pain and the occurrence of suicidal ideation. Pompili et al. [38] explained that individuals with higher educational achievement may be more prone to suicide risk when facing failures, public shame, and high premorbid functioning. However, several previous studies found the opposite results [32,39,40,41]. Belo et al. [39] reported that old-age people with high education have a better psychological adjustment concerning well-being, which is in consistent with our results. Oh et al.’s study [40] in Korea found that elderly people with low education level were associated with the risk of incident suicidality. Yen et al. [41] investigated community-dwelling elders in Taiwan and demonstrated that participants with lower educational level were more likely to experience severe depressive symptoms, thus indirectly leading to the occurrence of suicidal ideation. Yu et al. [32] conducted a population-based survey among older adults in Shandong, China, and also discovered a correlation between lower education level and suicidal ideation. Thus, further accurate studies are needed to understand the relationship between education level and psychological pain and the occurrence of suicidal ideation.

Significant association between the increase of psychological pain and the occurrence of suicidal ideation was found in this study, consistent with previous literature [21,42,43,44]. Psychological pain is proposed to be at the core of suicidality and has key effects on suicidal ideation and behaviors [43,45]. Unbearable psychological pain is the most often reported reason for suicidal behavior [21].

Several potential limitations should be mentioned regarding the present study. Firstly, convenience sampling was used to recruit participants from both offline and online sources, and this study did not consider underlying diseases and medication uses of the participants, which may cause sampling bias. Secondly, this study made an attempt to clarify the psychological pain and suicidal ideation of the elderly with different parental states. However, almost 3/4 of the elderly participants’ parental state was neither parent alive, which might cause some biases. Thirdly, the cross-sectional nature of this study may be considered a weakness, as no causal inferences can be drawn from the results. Despite these limitations, this study is the first to draw attention to the role of parental state on older adults’ mental health in China. The results might be a valuable references for the effective implementation of current “Elderly Health Promotion Action”, “Mental Health Promotion Action” [20], and future relevant research.

## 5. Conclusions

In China, the elderly with one parent alive experienced the lowest psychological pain and suicidal ideation, and those with both parents alive reported the highest psychological pain. Older adults who were single, divorced, or widowed; had higher education level, lower family income; suffered from two or more chronic diseases; and had no fully self-care ability were more likely to feel psychological pain and have suicidal ideation. Higher psychological pain significantly associated with the occurrence of suicidal ideation in the Chinese elderly. Thus, much more support (e.g., family care subsides, health education, psychological counseling service) should be offered to better prevent, detect, and control a mental health crisis in the Chinese elderly, especially for those with both parents alive. In essence, to provide a timely response and longer-lasting physical and mental integrated clinical support for family care in such a Confucian-influenced country, it is urgent to develop community-based clinical teams, offer home visits, temporary health-care or accessible E-health services, etc. Moreover, the associated factors above should be fully considered to develop targeted health interventions.

## Figures and Tables

**Table 1 ijerph-17-06399-t001:** Characteristics of the study participants.

Variables	Categories	*N* = 4622
Parental state	Neither alive	3414 (73.9)
	One alive	788 (17.0)
	Both alive	420 (9.1)
Age, mean ± SD		71.2 ± 7.8
Gender	Male	2284 (49.4)
	Female	2338 (50.6)
Marital status	Married	3129 (67.7)
	Others (single, divorced, and widowed)	1493 (32.3)
Place of residence	rural areas	2805 (60.7)
	main urban or urban-rural areas	1817 (39.3)
Education level	<Primary school	1942 (42.0)
	Primary school	1136 (24.6)
	Middle/high school	1192 (25.8)
	≥College	352 (7.6)
Family per-capita annual income (CHY)	≤15,000	1639 (35.5)
	15,000–30,000	1162 (25.1)
	30,000–45,000	883 (19.1)
	>45,000	938 (20.3)
Occupation before retirement	Farmer, worker, etc.	2657 (57.5)
	Self-employed	585 (12.7)
	Company employee	424 (9.2)
	Government employee	766 (16.6)
	Others	190 (4.1)
Number of chronic diseases	0	2219 (48.0)
	1	1252 (27.1)
	≥2	1151 (24.9)
Fully self-care ability	Yes	3265 (70.6)
	No	1357 (29.4)

Continuous variables are presented as mean and standard deviation (SD); categorical variables are presented as frequency (*n*) and percentage (%).

**Table 2 ijerph-17-06399-t002:** Psychological pain and suicidal ideation in participants with different parental states.

Variables	All Participants	Parental State			*F*/*χ*^2^	Pairwise Comparison
		Neither Alive (g1)	One Alive (g2)	Both Alive (g3)		
**Psychological pain**						
PSA sum-score	22.1 ± 9.5	22.5 ± 9.6	19.7 ± 7.6	23.3 ± 11.0	33.27 ***	g2 < g1 & g3
PSA-frequency	14.8 ± 6.6	15.0 ± 6.7	13.2 ± 5.2	15.7 ± 7.8	28.71 ***	g2 < g1 < g3
PSA-intensity	7.3 ± 3.4	7.5 ± 3.5	6.4 ± 2.8	7.6 ± 3.7	34.18 ***	g2 < g1 & g3
**Suicidal ideation**						
Wish to die					98.82 ***	g2 < g1 & g3
Moderate to severe	134 (2.9)	112 (3.3)	7 (0.9)	15 (3.6)		
Mild	1450 (31.4)	1164 (34.1)	147 (18.7)	139 (33.1)		
None	3038 (65.7)	2138 (62.6)	634 (80.5)	266 (63.3)		
Wish to live					64.65 ***	g2 < g1 & g3
None	152 (3.3)	117 (3.4)	11 (1.4)	24 (5.7)		
Mild	943 (20.4)	759 (22.2)	93 (11.8)	91 (21.7)		
Moderate to severe	3527 (76.3)	2538 (74.3)	684 (86.8)	305 (72.6)		
Reasons to live/die					47.26 ***	g2 < g1 & g3
More for dying	111 (2.4)	89 (2.6)	9 (1.1)	13 (3.1)		
Equal	1319 (28.5)	1042 (30.5)	155 (19.7)	122 (29.0)		
More for living	3192 (69.1)	2283 (66.9)	624 (79.2)	285 (67.9)		
Overall suicidal ideation					95.80 ***	g2 < g1 & g3
Yes	2061 (44.6)	1634 (47.9)	227 (28.8)	200 (47.6)		
No	2561 (55.4)	1780 (52.1)	561 (71.2)	220 (52.4)		

Continuous variables are presented as mean and standard deviation (SD); categorical variables are presented as frequency (*n*) and percentage (%). *** *p*-value < 0.001.

**Table 3 ijerph-17-06399-t003:** Linear regression model testing the associated factors of psychological pain.

Variables	*B*	*S.E.*	95% *CI* for *B*		*T*	*p*-Value
			Lower	Upper		
Parental state (vs. Neither alive)						
One alive	−0.95	0.38	−1.69	−0.20	6.18	0.013
Both alive	1.47	0.47	0.54	2.40	9.64	0.002
Age	−0.06	0.02	−0.09	−0.02	8.57	0.003
Gender (vs. Female)						
Male	−0.82	0.28	−1.36	−0.28	8.71	0.003
Marital status (vs. Others)						
Married	−1.02	0.30	−1.62	−0.43	11.30	0.001
Place of residence (vs. Main urban or urban-rural areas)						
Rural	1.30	0.34	0.63	1.97	14.56	0.000
Education level	0.55	0.16	0.22	0.87	11.03	0.001
Family income	−1.68	0.14	−1.94	−1.41	152.35	0.000
Occupation before retirement	0.17	0.13	−0.08	0.43	1.80	0.180
Number of chronic diseases (vs. 0)						
1	0.13	0.32	−0.49	0.75	0.17	0.681
≥2	2.35	0.33	1.71	3.00	50.83	0.000
Fully self-care ability (vs. No)						
Yes	−3.73	0.31	−4.34	−3.13	146.00	0.000

Goodness-of-fit test: *R^2^* = 0.122, adjusted *R^2^* = 0.120, *F* = 63.61, *p*-value < 0.05.

**Table 4 ijerph-17-06399-t004:** Binary logistic regression analysis testing the factors associated with suicidal ideation.

Variables	*OR*	95% *CI* for *OR*		*p*-Value
		Lower	Upper	
Parental state (vs. Neither alive)				
One alive	0.72	0.59	0.87	0.001
Both alive	1.25	1.00	1.56	0.053
Age	1.00	0.99	1.01	0.838
Gender (vs. Female)				
Male	0.90	0.79	1.02	0.104
Marital status (vs. Others)				
Married	0.57	0.49	0.66	0.000
Place of residence (vs. Main urban or urban-rural areas)				
Rural	1.17	1.00	1.38	0.055
Education level	1.28	1.18	1.38	0.000
Family income	0.69	0.64	0.73	0.000
Occupation before retirement	0.98	0.92	1.05	0.568
Number of chronic diseases (vs. 0)				
1	1.36	1.17	1.59	0.000
≥2	1.92	1.65	2.25	0.000
Fully self-care ability (vs. No)				
Yes	0.50	0.43	0.58	0.000

Goodness-of-fit test: Hosmer–Lemeshow *χ^2^* = 28.13, *p*-value > 0.05.

**Table 5 ijerph-17-06399-t005:** Association between psychological pain and suicidal ideation.

Variables	Overall Suicidal Ideation		*r*	*OR* (95% *CI*)
	Yes	No		
Psychological pain	27.7 ± 10.4	17.7 ± 5.5	0.44 ***	1.16 (1.14–1.17) ***
Pain frequency	18.4 ± 7.4	11.8 ± 3.8	0.42 ***	1.22 (1.20–1.24) ***
Pain intensity	9.2 ± 3.5	5.8 ± 2.4	0.43 ***	1.40 (1.37–1.44) ***

*** *p*-value < 0.001. Goodness-of-fit test: psychological pain, Hosmer–Lemeshow *χ^2^* = 9.98, *p*-value > 0.05; pain frequency, Hosmer–Lemeshow *χ^2^* = 12.85, *p*-value > 0.05; pain intensity, Hosmer–Lemeshow *χ^2^* = 11.37, *p*-value > 0.05. Regression model adjusted for parent state, age, gender, marital status, place of residence, education level, family per capital annual income (CHY), occupation before retirement, number of chronic diseases, and fully self-care ability.

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
