# Peer review of "Do Older Adults with Parent(s) Alive Experience Higher Psychological Pain and Suicidal Ideation? A Cross-Sectional Study in China"

_ijerph, 2020, doi:10.3390/ijerph17176399_

Round 1
Reviewer 1 Report
It is not frequent to find manuscripts with such a high level, as this. The authors approach the subject in a holistic way, using a clear methodology and supported by the large amount of data they were able to collect.
One specific point is highlighted in the methods section. More information should be incorporated into the selection of regression approach, in terms of how confounding factors were controlled, did you run correlation analysis among quantitative variables? did you perform any chi-square test for qualitative variables? How you build the models? Did you use backward elimination or forward elimination? how you check for the adjustment of the model to data? How you check for the linear relationship between quantitative independent variables with dependent variable?
I think that method section and statistical analysis must be supplemented with this information, in order to make the results more robust.

Author Response
We deeply appreciate the reviewers for all the valuable comments which are very helpful for us to improve the manuscript. We have revised the manuscript carefully according to each comment. All the corrections and revisions are highlighted in the revised manuscript. The following points are replies to each reviewer point by point.
Reply to reviewers’ report
Reviewer 1
Comments and Suggestions for Authors
It is not frequent to find manuscripts with such a high level, as this. The authors approach the subject in a holistic way, using a clear methodology and supported by the large amount of data they were able to collect.
RESPONSE
We really appreciate the reviewer’s high recognition of our study.
One specific point is highlighted in the methods section. More information should be incorporated into the selection of regression approach, in terms of how confounding factors were controlled, did you run correlation analysis among quantitative variables? did you perform any chi-square test for qualitative variables? How you build the models? Did you use backward elimination or forward elimination? how you check for the adjustment of the model to data? How you check for the linear relationship between quantitative independent variables with dependent variable?
I think that method section and statistical analysis must be supplemented with this information, in order to make the results more robust.
RESPONSE
We would like to thank the reviewer for the detailed comments on the statistical analysis section. According to the reminder, we have re-written the entire Statistical Analysis section and clarified the above questions mentioned by the reviewer. Please see page 3, lines 118-138.
Reviewer 2 Report
This study has a clear interesting and the results are in general novel. However, I wonder how the authors justified the findings that the older participants with both parents alive demonstrated the heaviest psychological pain? Can the authors elaborate more on this on introduction and discussion? Which are the implications of this finding?
Author Response
We deeply appreciate the reviewers for all the valuable comments which are very helpful for us to improve the manuscript. We have revised the manuscript carefully according to each comment. All the corrections and revisions are highlighted in the revised manuscript. The following points are replies to each reviewer point by point.
Reviewer 2
This study has a clear interesting and the results are in general novel. However, I wonder how the authors justified the findings that the older participants with both parents alive demonstrated the heaviest psychological pain? Can the authors elaborate more on this on introduction and discussion? Which are the implications of this finding?
RESPONSE
First of all, we would like to thank the reviewer for the recognition of our article and for the helpful comments. According to the reviewer’s advice, we have added relevant content to clarify this issue. Please see the highlighted text in the Introduction section (page 1, line 40, lines 44-45; page 2, line 46, lines 55-57) and the Discussion section (page 7, lines 226-232).
Reviewer 3 Report
The authors of this article intended to explore the psychological pain and suicide ideation among Chinese older adults base on whether both of their parents were alive. The Authors found that 1) older adults with both parents alive reported highest psychological pain; 2) older adults who were single, divorced, or widowed, live in rural areas, had higher education level, had lower family income, suffered from two or more chronic diseases, and had no self-care ability were more likely to experience psychological pain and suicidal ideation; 3) higher psychological pain was significantly associated with suicide ideation. It provided further support of the relationship between psychological pain and suicide ideation and investigated a rarely touched topic in our field. Specifically, it could aid our understanding on the mental health of older adults from families that’s being influenced by the Confucian culture.
Overall, this is a well written article. There are several concerns, however, that need to be addressed by the authors before I can recommend it for publication.
General comments:
- Please use “supported” instead of “confirmed” when addressing findings of previous studies. Most of the studies in our field were using the data to make inferences that we can not know for sure. The word “confirmed” gives people an impression that we are 100% sure of the inferences we made. Thus, “supported” would be a more accurate term to use when addressing findings of studies.
Intro:
- Please clearly state your hypothesis/hypotheses. If you do not have specific hypothesis/hypotheses, please indicate this is an exploratory study.
- It sounds like your second aim of the study should be separated into two aims “determine the associated factors of psychological pain and suicidal ideation” and “the relationship between psychological pain and suicidal ideation”
Methods:
- Consider change “self-completed” to “self report”.
- The authors measured suicidal ideation by asking participants to rate their wish to die, wish to live, and reasons for living and dying. They do not, however, ask participants directly about their wish to die by suicide or to kill themselves. There is a difference between a passive wish to die vs. an active desire for suicide. This is a major limitation that I believe requires the authors to 1) provide strong rationale for using these questions as a measure of suicidal ideation, 2) address this limitation in the discussion, and 3) use more tentative language in interpreting their findings due to this limitation.
Results:
- How were the predictor variables entered into the linear regression/logistic regression models? Were all predictor variables entered into the same model or were they ran separately? This was unclear in the tables and in the written results. Given the large sample size, I’m wondering if the analyses were overpowered and thus returning significant p-values for small/insignificant effects (i.e. false positives). It would be helpful for the authors to provide more information and discussion on the effect sizes that were found to support that their findings are clinically significant.
Discussion:
- The authors should provide additional discussion as to why participants with no parents alive reported higher psychological pain compared to people with one parents alive.
- Please provide some thoughts regarding why this particular study found elders with higher education level reported higher psychological pain and suicide ideation? Especially given that this contradicts findings from previous studies.
- I would find it helpful if the authors can provide some of the clinical/practical implications their findings.
Author Response
Reviewer 3
Comments and Suggestions for Authors
The authors of this article intended to explore the psychological pain and suicide ideation among Chinese older adults base on whether both of their parents were alive. The Authors found that 1) older adults with both parents alive reported highest psychological pain; 2) older adults who were single, divorced, or widowed, live in rural areas, had higher education level, had lower family income, suffered from two or more chronic diseases, and had no self-care ability were more likely to experience psychological pain and suicidal ideation; 3) higher psychological pain was significantly associated with suicide ideation. It provided further support of the relationship between psychological pain and suicide ideation and investigated a rarely touched topic in our field. Specifically, it could aid our understanding on the mental health of older adults from families that’s being influenced by the Confucian culture.
Overall, this is a well written article. There are several concerns, however, that need to be addressed by the authors before I can recommend it for publication.
RESPONSE
Thanks very much for the recognition of our article and all the valuable comments below.
General comments:
Comment 1#) Please use “supported” instead of “confirmed” when addressing findings of previous studies. Most of the studies in our field were using the data to make inferences that we can not know for sure. The word “confirmed” gives people an impression that we are 100% sure of the inferences we made. Thus, “supported” would be a more accurate term to use when addressing findings of studies.
RESPONSE
Done. Please see page 8, line 246.
Intro:
Comment 2#) Please clearly state your hypothesis/hypotheses. If you do not have specific hypothesis/hypotheses, please indicate this is an exploratory study.
RESPONSE
Done. Please see page 2, line 63.
Comment 3#) It sounds like your second aim of the study should be separated into two aims “determine the associated factors of psychological pain and suicidal ideation” and “the relationship between psychological pain and suicidal ideation”
RESPONSE
Done. Please see the Abstract section (page 1, lines 15-17) and the Introduction section (page 2, lines 65-68).
Methods:
Comment 4#) Consider change “self-completed” to “self report”.
RESPONSE
Done. Please see page 2, lines 72-73.
Comment 5#) The authors measured suicidal ideation by asking participants to rate their wish to die, wish to live, and reasons for living and dying. They do not, however, ask participants directly about their wish to die by suicide or to kill themselves. There is a difference between a passive wish to die vs. an active desire for suicide. This is a major limitation that I believe requires the authors to 1) provide strong rationale for using these questions as a measure of suicidal ideation, 2) address this limitation in the discussion, and 3) use more tentative language in interpreting their findings due to this limitation.
RESPONSE
We agree with the reviewer’s point and have made revision according to the reviewers’ specific suggestions.
Firstly, we have added relevant content to address the rationale for using the three structured questions to assess suicidal ideation. Please see the Materials and Methods section (page 3, lines 107-114). Secondly, in the Discussion section, we have mentioned this potential limitation. Please see page 7, lines 216-221.
Results:
Comment 6#) How were the predictor variables entered into the linear regression/logistic regression models? Were all predictor variables entered into the same model or were they ran separately? This was unclear in the tables and in the written results. Given the large sample size, I’m wondering if the analyses were overpowered and thus returning significant p-values for small/insignificant effects (i.e. false positives). It would be helpful for the authors to provide more information and discussion on the effect sizes that were found to support that their findings are clinically significant.
RESPONSE
We appreciate the reviewer for pointing out this issue and agree with the comment very much. In the revised manuscript, we have clarified this issue by re-written the entire Statistical Analysis section (page 3, lines 118-138). Also, we have added footnote in Table 5 (page 7, lines 200-202).
Discussion:
Comment 7#) The authors should provide additional discussion as to why participants with no parents alive reported higher psychological pain compared to people with one parents alive.
RESPONSE
We have added relevant content in the Discussion section (Page 7, lines 226-232).
Comment 8#) Please provide some thoughts regarding why this particular study found elders with higher education level reported higher psychological pain and suicide ideation? Especially given that this contradicts findings from previous studies.
RESPONSE
We have provided additional explanation in the Discussion section (page 8, lines 258-261).
Comment 9#) I would find it helpful if the authors can provide some of the clinical/practical implications their findings.
RESPONSE
We appreciate the suggestion and have provided some possible practical implications in the Conclusion section (Page 9, line 292-294).
Round 2
Reviewer 3 Report
Overall, the authors did a great job of addressing our comments. We are completely satisfied with the authors' revision base on comment 1# to comment 5# and comment 8#.
The authors’ revision on the statistical analysis section well demonstrated how they conducted their analyses (comment 6#). They, however, did not address our concerns regarding the power issues of the analyses. Given the large sample size, we are wondering if the analyses were overpowered and thus returning significant p-values for small/insignificant effects (i.e. false positives). It would be helpful for the authors to provide more information and discussion on the effect sizes that were found to support that their findings are clinically significant. Specifically, please provide R2 for liner regressions as well as η2 for ANOVA in your result section and provide an explanation/discussion about them in the discussion section (regarding whether they are small, medium, or large).
For comment 7#, the authors addressed our comments by striving to explain their findings regarding why elders with no parents alive reported higher psychological pain compared to those with one parent alive. It was a great explanation, but I believe it could be clearer. Using language like “compared to participants with one parent alive, participants with no parent alive…” could make the comparison more obvious.
For comment 9#
The authors provided a sentence in addressing the comment 9#. It was a great start, but we would like to hear more details from the authors. For example, how would the findings of this study inform the clinicians that are providing service to the elders from families being influenced by the Confucian culture?
Based on the aforementioned revision considerations, we believe a major revision is needed before it can be recommended for publication.
Author Response
We deeply appreciate the editor’s efficient work and the reviewer’s critical comments. According the reviewer’s comments and suggestions, we have revised the manuscript carefully. All the corrections and revisions are highlighted in the revised manuscript. The following points are replies to each reviewer point by point.
Reply to reviewers’ report
Reviewer 3
Overall, the authors did a great job of addressing our comments. We are completely satisfied with the authors' revision base on comment 1# to comment 5# and comment 8#.
RESPONSE
Thanks very much for the recognition of our revision. For the three comments with unsatisfied modification in the first round, we have made further changes according to your suggestions.
(1) The authors’ revision on the statistical analysis section well demonstrated how they conducted their analyses (comment 6#). They, however, did not address our concerns regarding the power issues of the analyses. Given the large sample size, we are wondering if the analyses were overpowered and thus returning significant p-values for small/insignificant effects (i.e. false positives). It would be helpful for the authors to provide more information and discussion on the effect sizes that were found to support that their findings are clinically significant. Specifically, please provide R2 for liner regressions as well as η2 for ANOVA in your result section and provide an explanation/discussion about them in the discussion section (regarding whether they are small, medium, or large).
RESPONSE
We appreciate the reviewer’s valuable and detailed suggestion. We have made revision accordingly.
In the Results section, we have provided η2 for ANOVA tests (page 5, lines 163-164), R2 for liner regression model (page 5, lines 181-182; page 6, line 184), and Hosmer-Lemeshow c2 for logistic regression models (page 6, lines 192-193; page 7, line 195, lines 202-204, and lines 206-208).
In the Discussion section, we have provided explanation on the small contribution of parental state on psychological pain (page 7, page 233-234).
(2) For comment 7#, the authors addressed our comments by striving to explain their findings regarding why elders with no parents alive reported higher psychological pain compared to those with one parent alive. It was a great explanation, but I believe it could be clearer. Using language like “compared to participants with one parent alive, participants with no parent alive…” could make the comparison more obvious.
RESPONSE
Thanks for the valuable comment. According to the reviewer’s suggestion, we have revised the sentence as follows (page 8, lines 239-242):
As for elderly participants without parents alive, since relieve burden for caring their parents, they appear less suicidal ideation. However, compared to participants with one parent alive, they could not benefit from good parent-child relationship. This finding might affirm the continued importance of cultural norms of elder care.
(3) For comment 9#, The authors provided a sentence in addressing the comment 9#. It was a great start, but we would like to hear more details from the authors. For example, how would the findings of this study inform the clinicians that are providing service to the elders from families being influenced by the Confucian culture?
Based on the aforementioned revision considerations, we believe a major revision is needed before it can be recommended for publication.
RESPONSE
Many thanks for the constructive comment. We have provided detailed information on the potential implication of our findings in the Conclusion section (page 9, lines 303-309). As follows:
Thus, much more support (e.g. family care subsides, health education, psychological counseling service) should be carried out to better prevent, detect, and control mental health crisis of Chinese elderly, especially for those with both parents alive. In essence, to provide timely-response, longer-lasting physical and mental integrated clinical support for family care in such a Confucian-influenced country, it is urgent to develop community-based clinical team, offer home visit, temporary health-care or accessible E-health service, etc. Moreover, the associated factors above should be fully considered to develop targeted health interventions.
